# The Effect of the Intrinsic Dimension on the Generalization of Quadratic Classifiers

**Fabian Latorre, Leello Dadi, Paul Rolland and Volkan Cevher**
Laboratory for Information and Inference Systems, EPFL, Switzerland.
`{first}.{last}@epfl.ch`

## Abstract

It has been recently observed that neural networks, unlike kernel methods, enjoy a reduced sample complexity when the distribution is isotropic (i.e., when the covariance matrix is the identity). We find that this sensitivity to the data distribution is not exclusive to neural networks, and the same phenomenon can be observed on the class of quadratic classifiers (i.e., the sign of a quadratic polynomial) with a nuclear-norm constraint. We demonstrate this by deriving an upper bound on the Rademacher Complexity that depends on two key quantities: (i) the intrinsic dimension, which is a measure of isotropy, and (ii) the largest eigenvalue of the second moment (covariance) matrix of the distribution. Our result improves the dependence on the dimension over the best previously known bound and precisely quantifies the relation between the sample complexity and the level of isotropy of the distribution.

## 1 Introduction

We revisit the problem of supervised classification using quadratic features of the data. We do so to highlight the influence of properties of data distrbution on the generalization error. Most of the existing results on this error only use a bound on the support of the distribution. By leveraging results from matrix concentration, we show an improved bound that uses more refined properties of the data distribution, like the second moment matrix.

The use of the second moment matrix in the error bound shows that the intrinsic dimension of the data distribution plays an important role. This is of particular interest because it is widely believed that real-world data distributions have nice properties that allow classifiers, namely neural networks, to avoid the worst-case sample complexities predicted by generalization bounds Jiang* et al. (2020).

Indeed, assumptions like the manifold hypothesis, which state that the data lies on lower dimensional embedded manifold, are often made to explain the practical success of some generative methods. A recent paper by Pope et al. (2021) computes estimates of this *true dimensionality* of common machine learning datasets and shows that they are much lower than the ambient dimension of the pixel space $[0, 1]^d$. It is therefore important that properties of the data distribution, going beyond simple bounds on the support, intervene in the study of generalization.

This influence of intrinsic dimension on generalization has been recently observed in the context of differentiating neural networks and from their kernel approximations, like the neural tangent kernel Jacot et al. (2018) or random feature models Yehudai and Shamir (2019). In particular, Ghorbani et al. (2020) observe that neural networks seem to require fewer samples than kernel methods to learn when the data distribution is isotropic.

We show that a similar phenomenon occurs in the simpler setting of quadratic classifiers, which leads to a better understanding of the causes. An improvement in sample complexity on isotropic data

35th Conference on Neural Information Processing Systems (NeurIPS 2021).

distributions can be proved when comparing nuclear-norm constrained quadratic classifiers and the corresponding kernel method (Frobenius norm constrained classifiers).

The study of quadratic classifiers can serve as an important first step in understanding how neural networks take advantage of the intrinsic dimension to learn with fewer samples Du and Lee (2018); Bai and Lee (2020). The nuclear-norm constraint is a natural one to study in this context. Indeed, when applying weight-decay (or $\ell_2$-regularization) on a single-hidden layer neural network with quadratic activations, the regularization is in effect encouraging a low nuclear norm of the coefficient matrix of the quadratic polynomial.

A better understanding of quadratics is also a worthwhile goal in its own right: complex architectures like those in Jayakumar et al. (2020) use quadratics as building blocks, attention layers Vaswani et al. (2017), which have seen great success in language processing tasks, are multiplicative interactions.

For these reasons, we present theoretical and practical developments of nuclear-norm regularization for quadratic classification. We summarize our contributions as follows

**Rademacher complexity bounds.**   We present a new bound on the Rademacher complexity of quadratic classifiers with a nuclear norm constraint c.f. Theorem 1. It improves upon the previously known bound, implied by the results by Kakade et al. (2012), by up to a square-root factor of the dimension, depending on the distribution of the data c.f. Lemma 4.

As a consequence of our bound, we draw attention to a clear difference between the complexity of nuclear-norm constrained and Frobenius-norm constrained quadratic classifiers. When the input data distribution is nearly-isotropic, the former enjoys a reduced dependency on the dimension. In contrast, the complexity of Frobenius-norm constrained classifiers has the same dependency on the dimension, independently of how isotropic the input data distribution is (Corollary 2).

This observation motivates the use of *data whitening* pre-processing steps, which are commonly used in practice: such transformation might bring the *second-order moment (covariance) matrix* of the distribution close to the identity matrix and thus to nearly-isotropicity.

**Computable generalization bounds.**   The refined Rademacher complexity bound that we obtain depends on the often unknown second-order moment of the distribution, rather than simple bounds on the diameter of the support as in (Kakade et al., 2012). Even though useful in theory, it is desirable in practice to obtain bounds that can be computed from a sample. We overcome this difficulty in Theorem 3, where we provide high-probability computable generalization error bounds for nuclear-norm constrained quadratic classifiers.

**Experiments.**  We illustrate our theoretical results on synthetic data. We show how the isotropy of the input distribution plays a major role in the generalization properties of quadratic classifiers. As the dimension increases and the sample size remains proportional to it, we observe a constant generalization gap for the nuclear-norm constrained classifier. In contrast, for SVMs, the gap grows at a predicted $\sqrt{d}$ rate. In the case of anisotropic distributions, we observe similar performance for both regularization schemes.

## 1.1   Related work

Kakade et al. (2012) provides generalization error bounds for the more general problem of learning a linear classifier over matrices. An upper bound for quadratic classifiers with a nuclear norm constraint can be derived as a consequence of their results c.f. Corollary 1. To the best of our knowledge, it is the only known bound for the hypothesis class we study, and thus the one we compare to. Our analysis improves the dependency on the dimension. See subsection 2.1 for a technical discussion.

Because of the generality of the results in Kakade et al. (2012), it is only natural that the implied bound in some particular case is not the tightest. We precisely give a step towards tight complexity estimates for classification with quadratic polynomials. We look on our results as relevant, given the simplicity and widespread use of linear learning over features.

Wimalawarne et al. (2016) study linear classifiers over higher-order tensor spaces, using constraints on generalized notions of the nuclear norm. The problem we study is thus a particular case. Generalization error bounds via Rademacher complexity are provided, but they apply only under a highly restrictive assumption: the entries of the tensor are independent standard normal random variables. In contrast we only require a boundedness condition.

Srebro et al. (2005); Srebro and Shraibman (2005) develop the theory of nuclear-norm regularization for matrix completion. Bounds on the Rademacher complexity of the class of matrices with bounded nuclear-norm are obtained in Srebro (2005). However, in this setting matrices are understood as mappings from an index pair to a value, and the generalization error measures how well the missing entries of the target matrix can be predicted. In contrast, in our setting the matrix corresponds to the coefficients of a quadratic polynomial, so the bounds are not comparable.

Pontil and Maurer (2013) study nuclear-norm regularization in the context of multi-task learning (Caruana, 1997). Rademacher complexity bounds are obtained for nuclear-norm constrained multi-task classifiers. In this setting, each row of the matrix corresponds to a linear classifier for a different task, and each task corresponds to a different distribution over data-label pairs.

Matrices with bounded trace norm also have been studied by Amit et al. (2007) and Yu et al. (2014) in the related *multi-class classification* and *multi-label learning* setting, respectively. Yu et al. (2014) remark essential differences between Rademacher complexity bounds of nuclear- and Frobenius-norm constrained linear classifiers, similar to our conclusion in Corollary 2. In all such problems, however, the setting is not comparable to ours: the matrix acts as a linear mapping of the sample, rather than as a quadratic. Thus, the analysis is not analogous and requires a different set of tools in our case.

The analysis in Yu et al. (2014) is closest in spirit to ours, as their bound also depends on the intrinsic dimension of the distribution, and the largest eigenvalue of the second moment (covariance) matrix. However, we go the extra mile and achieve bounds that can be computed from the sample at hand (Theorem 3), as the true second moment (covariance) matrix is usually unknown.

The papers Du and Lee (2018); Bai and Lee (2020) establish a similar result as ours for a different norm for a non-convex parametrization of quadratic polynomials. The reparametrization consits of writing the coefficient matrix as a sum of $m$ rank 1 matrices which facilitates analogies to neural networks. It is straighforward to see that their studied norm $\|\cdot\|_{2,4}$ is, in essence, a way of upper bounding the nuclear-norm by using the reparametrization.

## 2 Rademacher complexity bounds

**Notation.** Throughout this section $\mathbf{x} \in \mathbb{R}^d$ is a random variable with distribution $\mu$, and $\mathbf{X}_n = (\mathbf{x}_1, \dots \mathbf{x}_n)$ is a sample of i.i.d. random variables drawn from $\mu$. The *second moment matrix* of $\mu$ is denoted by $\Sigma := \mathbb{E}[\mathbf{x}\mathbf{x}^T]$. Note that, for centered (mean zero) random variables, this notion coincides with the *covariance matrix*. For a square symmetric matrix $\mathbf{A}$ we denote with $\|\mathbf{A}\|_F, \|\mathbf{A}\|_2$ and $\|\mathbf{A}\|_{\mathrm{tr}}$ its *Frobenius-norm*, *spectral-norm* and *nuclear-norm*, respectively. The notation $\lesssim$ stands for *less than or equal*, but hides constants independent of the dimension or number of samples. The notation $x \approx y$ means that there exist constants $c, C > 0$ such that $cy \leq x \leq Cy$.

We consider a binary classifier obtained from a homogeneous quadratic polynomial. Such a function can be parametrized as $f_{\mathbf{A}}(\mathbf{x}) := \mathbf{x}^T \mathbf{A}\mathbf{x}$, where $\mathbf{A}$ is a square symmetric matrix containing the coefficients of the monomials. In order to control the *complexity* of a quadratic polynomial we choose a matrix norm $\|\cdot\|$ and consider only elements in a constrained set:

$$\mathcal{Q}_{\|\cdot\|,\lambda} := \{f_{\mathbf{A}}(\mathbf{x}) = \mathbf{x}^T \mathbf{A}\mathbf{x} : \|\mathbf{A}\| \leq \lambda\}, \qquad \mathcal{Q}_{\|\cdot\|} := \mathcal{Q}_{\|\cdot\|,1} \tag{1}$$

We quantify the complexity of such function classes using the classical notion of *Rademacher complexity*. It is well known that high probability generalization error bounds can be obtained in terms of this quantity (Koltchinskii and Panchenko, 2002; Bartlett and Mendelson, 2003). For this reason, we focus only on deriving upper bounds on this complexity measure.

**Definition 1** (Rademacher complexity). *Let $\sigma$ be uniformly distributed over the set $\{-1, 1\}^n$ and let $\mathbf{X}_n = [\mathbf{x}_1, \dots, \mathbf{x}_n] \subseteq \mathbb{R}^d$ be an i.i.d. sample drawn according to $\mu$. For a class of functions $\mathcal{F} : \mathbb{R}^d \to \mathbb{R}$ we define the empirical Rademacher complexity and the Rademacher complexity (with respect to $\mu$) of $\mathcal{F}$, respectively, as:*

$$\hat{\mathcal{R}}(\mathcal{F}; \mathbf{X}_n) := \mathbb{E}_\sigma \left[ \sup_{f \in \mathcal{F}} \frac{1}{n} \sum_{i=1}^n \sigma_i f(\mathbf{x}_i) \right], \qquad \mathcal{R}_{n,\mu}(\mathcal{F}) = \mathbb{E}[\hat{\mathcal{R}}(\mathcal{F}; \mathbf{X}_n)] \tag{2}$$

Our bounds depend on the distribution through its *intrinsic dimension* (Tropp, 2015, Section 7), which measures how much the probability density concentrates near low-dimensional subspaces.

**Definition 2.** *The intrinsic dimension of a distribution $\mu$ is the ratio $1 \leq r(\Sigma) := \mathrm{tr}(\Sigma)/\|\Sigma\|_2 \leq d$.*

We are now ready to state our main results about the Rademacher complexity of homogeneous quadratic polynomials with nuclear norm constraint:

**Theorem 1.** *Let $\mathbf{x} \in \mathbb{R}^d$ such that $\|\mathbf{x}\|_2^2 \lesssim \mathbb{E}\|\mathbf{x}\|_2^2$ almost surely and suppose $n \gtrsim r(\Sigma) \log d$. It holds that*

$$\mathcal{R}_{n,\mu}(\mathcal{Q}_{\|\cdot\|_{\mathrm{tr}},\lambda}) \lesssim \lambda \sqrt{\frac{r(\Sigma) \log d}{n}} \|\Sigma\|_2 \tag{3}$$

Now we proceed to prove Theorem 1. First, we only focus on the class $\mathcal{Q}_{\|\cdot\|}$ corresponding to the unit nuclear-norm ball. This is justified by well-known technical result Lemma 1, whose proof is included for completeness in Appendix D.

**Lemma 1.** $\mathcal{R}_{n,\mu}(\mathcal{Q}_{\|\cdot\|,\lambda}) \leq \lambda \mathcal{R}_{n,\mu}(\mathcal{Q}_{\|\cdot\|})$.

The backbone of Theorem 1 is Lemma 3, which relates the Rademacher complexity of a class of functions to concentration of empirical means to expectations with respect to the *dual norm*. It makes use of the technical Lemma 2, whose proof is included for completeness in Appendix B.

**Lemma 2.** *For all even $n \in \mathbb{N}$, it holds that $\sum_{k=0}^{n} |2k - n| \binom{n}{k} < \sqrt{n} 2^n$*

**Lemma 3.** *Denote by $\|\cdot\|_*$ the dual norm of $\|\cdot\|$. Define $M_k := \mathbb{E} \|\Sigma_k - \Sigma\|_*$, $\Sigma_k := \frac{1}{k} \sum_{i=1}^{k} \mathbf{x}_i \mathbf{x}_i^T$ The Rademacher complexity of the class $\mathcal{Q}_{\|\cdot\|}$ can be upper bounded as follows:*

$$\mathcal{R}_{n,\mu}(\mathcal{Q}_{\|\cdot\|}) \leq \frac{1}{n 2^{n-1}} \sum_{k=1}^{n} k \binom{n}{k} M_k + \frac{\|\Sigma\|_*}{\sqrt{n}} \tag{4}$$

*Proof.* We first compute an upper bound on the empirical Rademacher complexity. The result will follow after taking expectation of the bound over the sample $\mathbf{X}_n = [\mathbf{x}_1, \ldots, \mathbf{x}_n]$. By definition of the dual norm, using the basic algebraic identity $\mathbf{x}^T \mathbf{A} \mathbf{x} = \langle \mathbf{A}, \mathbf{x}\mathbf{x}^T \rangle$ we have

$$\hat{\mathcal{R}}(\mathcal{Q}_{\|\cdot\|}; \mathbf{X}_n) = \frac{1}{n} \mathbb{E}_\sigma \sup_{\|\mathbf{A}\| \leq 1} \left\langle \mathbf{A}, \sum_{i=1}^{n} \sigma_i \mathbf{x}_i \mathbf{x}_i^T \right\rangle = \frac{1}{n} \mathbb{E}_\sigma \left\| \sum_{i=1}^{n} \sigma_i \mathbf{x}_i \mathbf{x}_i^T \right\|_* \tag{5}$$

We now compute the expectation in eq. 5. There is a bijection between the possible configurations of the Rademacher variable $\sigma \in \{-1, 1\}^n$ and the *power set* of $[n]$, namely $\sigma \mapsto \{i \in [n] : \sigma_i = 1\}$. This allows us to write eq. 5 as:

$$\hat{\mathcal{R}}(\mathcal{Q}_{\|\cdot\|}; \mathbf{X}_n) = \frac{1}{n} \mathbb{E}_\sigma \left\| \sum_{i=1}^{n} \sigma_i \mathbf{x}_i \mathbf{x}_i^T \right\|_* = \frac{1}{n 2^n} \sum_{B \subseteq [n]} \underbrace{\left\| \sum_{i \in B} \mathbf{x}_i \mathbf{x}_i^T - \sum_{i \in B^c} \mathbf{x}_i \mathbf{x}_i^T \right\|_*}_{:=D_B} \tag{6}$$

Let $\Sigma_B := |B|^{-1} \sum_{i \in B} \mathbf{x}_i \mathbf{x}_i^T$. Using the triangle inequality, we can bound $D_B$ as:

$$D_B \leq |B| \|\Sigma_B - \Sigma\|_* + \big||B| - |B^c|\big| \|\Sigma\|_* + |B^c| \|\Sigma_{B^c} - \Sigma\|_* \tag{7}$$

To obtain a bound on the Rademacher complexity, we need now sum over $B \subseteq [n]$ the terms on the right hand side of eq. 7, and take expectation with respect to the sample $\mathbf{X}_n$. First, we will deal with the sum of the second term in eq. 7, as it is actually a deterministic value. We can sum over $B \subseteq [n]$ by grouping together subsets $B$ of the same cardinality $|B| = k$. We obtain:

$$\sum_{B \subseteq [n]} \big||B| - |B^c|\big| \|\Sigma\|_* = \sum_{k=0}^{n} \binom{n}{k} |2k - n| \|\Sigma\|_* \leq \sqrt{n} 2^n \|\Sigma\|_* \tag{8}$$

where the last inequality follows from Lemma 2.

Finally, we compute the expectation of the sum over $B \subseteq [n]$ of the first and third term in eq. 7. After taking the sum, both terms become equal by symmetry. It suffices to bound the sum of the first term.

Notice that because the variables $\mathbf{x}_1, \ldots, \mathbf{x}_n$ are i.i.d., the distribution of $\Sigma_B$ depends only on the size of the set $B$. Using the same counting argument as in eq. 8 we arrive at:

$$
\begin{aligned}
\mathbb{E} \sum_{B \subseteq [n]} |B| \|\Sigma_B - \Sigma\|_* &= \sum_{B \subseteq [n]} |B| \mathbb{E} \|\Sigma_B - \Sigma\|_* \\
&= \sum_{k=1}^n k \binom{n}{k} \underbrace{\mathbb{E} \|\Sigma_k - \Sigma\|_*}_{=M_k}
\end{aligned}
\tag{9}
$$

Combining the bounds in eq. 8 and eq. 9, and dividing by $n2^n$ we obtain the result. $\qquad \square$

Note that the proof of Lemma 3 follows from technical arguments but the final result is not to be found in the literature, in this form or a similar one. In particular, it is completely unrelated to the result by Vershynin (2011) with which it only shares the fairly trivial split of Rademacher random variables preceding eq. 6.

Lemma 3 provides a way to derive Rademacher complexity bounds from a bound on the *expected deviations* $M_k$ defined in Lemma 3, and might be of independent interest. In the particular case where the norm in consideration is the nuclear-norm, this lemma will be used to establish Theorem 1 as a simple application of a well-known non-asymptotic bound for the convergence of the empirical second moment (covariance) matrix to the true second moment (covariance) matrix.

*Proof of Theorem 1.* Recall that the dual norm of the nuclear-norm is the spectral-norm. The value of $M_k$ in Lemma 3 measures the average deviation of the *empirical second moment matrix* $\Sigma_k$ to the true $\Sigma$, in spectral-norm. Our assumption that $\|\mathbf{x}\|_2^2 \lesssim \mathbb{E}\|\mathbf{x}\|_2^2$ almost surely, implies the concentration result (Vershynin, 2018, Theorem 5.6.1), which concludes that

$$
M_k \lesssim \left( \sqrt{\frac{r(\Sigma) \log d}{k}} + \frac{r(\Sigma) \log d}{k} \right) \|\Sigma\|_2
\tag{10}
$$

Plug this in eq. 4, and use the bound $\sqrt{k} \leq \sqrt{n}$ for $k \leq n$ to obtain the inequality

$$
\mathcal{R}_{n,\mu}(\mathcal{Q}_{\|\cdot\|_{\mathrm{tr}}}) \lesssim \left( \sqrt{\frac{r(\Sigma) \log d}{n}} + \frac{r(\Sigma) \log d}{n} \right) \|\Sigma\|_2
\tag{11}
$$

By assumption $n \gtrsim r(\Sigma) \log d$, so that the first term in eq. 11 is the largest. The second term is of smaller order and thus ends up hidden by the notation $\lesssim$. We conclude

$$
\mathcal{R}_{n,\mu}(\mathcal{Q}_{\|\cdot\|_{\mathrm{tr}}}) \lesssim \sqrt{\frac{r(\Sigma) \log d}{n}} \|\Sigma\|_2
\tag{12}
$$

Invoking Lemma 1 we obtain the desired result. $\qquad \square$

## 2.1 Improvement upon previous work

We now show how our derived upper bound improves over the current best known bound by Kakade et al. (2012). To the best of our knowledge, the Rademacher complexity of quadratic classifiers with a nuclear-norm constraint has not been previously analyzed in a *direct manner*, as we do. Instead, the only existing bound (Corollary 1) appears as a particular case of Theorem 2.

**Theorem 2** (Kakade et al. (2012) page 1876)**.** *Let*

$$
\mathcal{G}_{\|\cdot\|_{\mathrm{tr}}, \lambda} := \{ g_{\mathbf{A}}(\mathbf{X}) := \langle \mathbf{A}, \mathbf{X} \rangle : \|\mathbf{A}\|_{\mathrm{tr}} \leq \lambda \}
\tag{13}
$$

*be the class of nuclear-norm constrained linear functions over square $d \times d$ matrices. Let $\mu$ be a distribution supported on $\mathcal{X} \subseteq \mathbb{R}^{d \times d}$. It holds that:*

$$
\mathcal{R}_{n,\mu}(\mathcal{G}_{\|\cdot\|_{\mathrm{tr}}, \lambda}) \lesssim \lambda X_\infty \sqrt{\frac{\log d}{n}}, \quad X_\infty = \sup_{\mathbf{X} \in \mathcal{X}} \|\mathbf{X}\|_2
$$

**Corollary 1.** *Let* $\mathbf{x} \in \mathcal{X} \subseteq \mathbb{R}^d$ *be a random variable with distribution* $\mu$. *The Rademacher complexity of the class in eq. 1 can be bounded as*

$$\mathcal{R}_{n,\mu}(\mathcal{Q}_{\|\cdot\|_{\mathrm{tr}},\lambda}) \lesssim \lambda x_\infty \sqrt{\frac{\log d}{n}}, \quad x_\infty := \sup_{\mathbf{x} \in \mathcal{X}} \|\mathbf{x}\|_2^2$$

Corollary 1 is a consequence of the fact that a function of the form $f_{\mathbf{A}}(\mathbf{x}) = \mathbf{x}^T \mathbf{A} \mathbf{x}$ can be written as a linear classifier on matrices, $g_{\mathbf{A}}(\mathbf{X}) = \langle \mathbf{A}, \mathbf{X} \rangle$, where $\mathbf{X} = \mathbf{x}\mathbf{x}^T$. In this case, it is easy to see that $X_\infty = \sup_{\mathbf{x} \in \mathcal{X}} \|\mathbf{x}\mathbf{x}^T\| = \sup_{\mathbf{x} \in \mathcal{X}} \|\mathbf{x}\|_2^2 =: x_\infty$. The only difference with our bound in Theorem 1 is that the term $x_\infty$ appears in place of $\sqrt{r(\Sigma)}\|\Sigma\|_2$.

In order to understand the difference between the two bounds, we turn to the analysis of the quotient between the bound in Corollary 1 and our bound Theorem 1. In Lemma 4 we show that this quotient can differ drastically, depending on the distribution.

**Remark 1.** *The variables* $X_\infty$ *and* $x_\infty$ *defined respectively in Theorem 2 and Corollary 1 respectively, correspond to the supremum of a random variable. Because the Rademacher complexity arises as an expectation, it is clear that such quantities can be (and should be) replaced by the closely related measure-theoretic notion of essential supremum (denoted by* ess sup*): the least upper bound that holds almost surely. In this way the bounds are only tighter, and we believe this was the true intended definition by Kakade et al. (2012). In the following we will compare our bound to this tighter, modified bound.*

**Lemma 4.** *Let* $\mathbf{x}$ *be a random variable supported on a set* $\mathcal{X} \subseteq \mathbb{R}^d$, *and such that* $\|\mathbf{x}\|_2^2 \lesssim \mathbb{E}\|\mathbf{x}\|_2^2$ *almost surely, then:*

$$\sqrt{r(\Sigma)} \lesssim \frac{\mathrm{ess\,sup}_{\mathbf{x} \in \mathcal{X}} \|\mathbf{x}\|_2^2}{\sqrt{r(\Sigma)}\|\Sigma\|_2} \lesssim \sqrt{r(\Sigma)} \tag{14}$$

*Proof.* By definition of essential supremum it holds that $\mathbb{E}\|\mathbf{x}\|_2^2 \leq \mathrm{ess\,sup}_{\mathbf{x} \in \mathcal{X}} \|\mathbf{x}\|_2^2$. Further, our assumption clearly implies that $\mathrm{ess\,sup}_{\mathbf{x} \in \mathcal{X}} \|\mathbf{x}\|_2^2 \lesssim \mathbb{E}\|\mathbf{x}\|_2^2$. The identity $\mathbb{E}\|\mathbf{x}\|_2^2 = \mathrm{tr}(\Sigma)$ and Definition 2 imply the result. $\square$

In summary, Lemma 4 shows that the baseline bound of Kakade et al. (2012) is larger by a square-root factor of the intrinsic dimension of the distribution (modulo global constants), compared to our bound in Theorem 1. Such factor ranges between 1 and the square root of the ambient dimension.

Precisely, when the intrinsic dimension of the distribution is equal to the ambient dimension, our bound enjoys a reduced dimension complexity. This is the case, for example, for isotropic distributions i.e., distributions such that their second moment matrix is the identity matrix. The dependency of the Rademacher complexity on the intrinsic dimension of the distribution is not revealed by the more general proof of Theorem 2 (Kakade et al., 2012).

**Remark 2.** *The logarithmic term in Theorem 1 can be removed under the more restrictive dimension-independent L-subgaussianity assumption (Mendelson and Zhivotovskiy, 2018).*

## 3 Computable Generalization error bounds

Let $\mathbf{y} = (y_1, \ldots, y_n) \in \{-1, 1\}^n$ be the labels associated with the data sample, and let

$$L(f) := \mathbb{P}\{\mathrm{sign}(f(\mathbf{x})) \neq \mathbf{y}\} \tag{15}$$

$$\hat{L}(f; \mathbf{X}_n) := \frac{1}{n} \sum_{i=1}^{n} \min(1, \max(0, 1 - y_i f(\mathbf{x}_i))) \tag{16}$$

be the *missclassification probability* and the *empirical margin loss* of the classifier $f$, respectively. It is well-known (Mohri et al., 2018, Theorem 5.8.) that with probability at least $1 - \delta$, for all $f \in \mathcal{Q}_{\|\cdot\|_{\mathrm{tr}},\lambda}$:

$$L(f) \lesssim \hat{L}(f; \mathbf{X}_n) + \mathcal{R}_{n,\mu}(\mathcal{Q}_{\|\cdot\|_{\mathrm{tr}},\lambda}) + \sqrt{\frac{\log \frac{1}{\delta}}{2n}} \tag{17}$$

This bound, together with the results in section 2, allow high probability uniform bounds on the misclassification error of a nuclear-norm constrained quadratic classifier.

However, the bound derived in this way is not actually computable: the Rademacher complexity bound in Theorem 1 depends on the second moment matrix of the distribution, which is unknown in practical applications. In the rest of this section we will overcome this drawback.

Rewriting our bound in eq. 3 as:

$$\mathcal{R}_{n,\mu}(\mathcal{Q}_{\|\cdot\|_{\mathrm{tr}}}) \lesssim \sqrt{\frac{\mathrm{tr}\,\Sigma \|\Sigma\|_2 \log d}{n}} \tag{18}$$

we observe that we need to estimate the trace and the largest eigenvalue of the second order moment matrix. Our hope is that the empirical estimators, based on the empirical second moment matrix $\Sigma_n$, will provide a good approximation of the true values.

We arrive at the following high probability bound on the Rademacher complexity in eq. 18, which is readily computable from samples and a bound on the diameter of the support of the distribution:

**Theorem 3.** *Suppose that $\|\mathbf{x}\|_2^2 \lesssim \mathbb{E}\|\mathbf{x}\|_2^2$ almost surely. This implies that $\|\mathbf{x}\|_2^2 \leq B$ for some $B > 0$. Define*

$$K_1 := \frac{\log d}{n - \sqrt{nd(\log d + \log \frac{1}{\delta})}} \qquad K_2 := \frac{\sqrt{2\log \frac{1}{\delta}}\log d}{2n(\sqrt{n} - \sqrt{d(\log d + \log \frac{1}{\delta})})} \tag{19}$$

*Let $\delta > 0$ and $f_\mathbf{A} \in \mathcal{Q}_{\|\cdot\|_{\mathrm{tr}},\lambda}$. Provided $n \gtrsim d(\log d + \log \frac{1}{\delta})$, with probability at least $1 - 3\delta$ it holds that*

$$\mathcal{R}_{n,\mu}(\mathcal{Q}_{\|\cdot\|_{\mathrm{tr}}}) \lesssim \underbrace{\sqrt{K_1 \,\mathrm{tr}\,\Sigma_n\|\Sigma_n\|_2 + BK_2\|\Sigma_n\|_2}}_{:=M(n,d,\delta)} \tag{20}$$

*Moreover, with probability at least $1 - 4\delta$, uniformly for all $f_\mathbf{A} \in \mathcal{Q}_{\|\cdot\|_{\mathrm{tr}},\lambda}$ it holds that*

$$L(f_\mathbf{A}) \lesssim \hat{L}(f_\mathbf{A}; \mathbf{X}_n) + \lambda M(n,d,\delta) + \sqrt{\frac{\log \frac{1}{\delta}}{2n}} \tag{21}$$

*Proof.* See Appendix C. $\qquad\qquad\qquad\qquad\qquad\qquad\qquad\qquad\qquad\qquad\qquad\qquad\qquad\qquad$ $\square$

## 4 Frobenius vs Nuclear-norm constraint

Perhaps the most common way to use quadratic features is to use a support vector machine (SVM) with the quadratic kernel $K(\mathbf{x}, \mathbf{y}) := \langle \mathbf{x}, \mathbf{y} \rangle^2$. Lemma 5, which is folklore in the kernel methods literature, precisely states that the RKHS norm constraint is equivalent to a Frobenius-norm constraint on the matrix of coefficients $\mathbf{A}$ of the underlying quadratic polynomial. Its proof is included for completeness in Appendix D.

**Lemma 5.** *Let $\mathbb{H}$ be the Reproducing Kernel Hilbert Space associated to the symmetric, positive semidefinite polynomial kernel $K(\mathbf{x}, \mathbf{y}) = \langle \mathbf{x}, \mathbf{y} \rangle^2$, and denote its induced norm by $\|\cdot\|_{\mathbb{H}}$. Then $f \in \mathbb{H}$ if and only if there exists a symmetric matrix $\mathbf{A}$ such that $f(\mathbf{x}) = \mathbf{x}^T \mathbf{A} \mathbf{x}$ and $\|f\|_{\mathbb{H}} = \|\mathbf{A}\|_F$.*

For this reason, we now turn to compare the qualities of nuclear-norm and Frobenius-norm constrained quadratic classifiers. As a consequence of our derived bound (eq. 3), we uncover a fundamental difference between both regularization schemes (Corollary 2): as the dimension increases, the growth rate of the complexity of nuclear-norm constrained quadratics strongly depends on the intrinsic dimension of the distribution. In contrast, that of Frobenius-norm constrained quadratics is insensitive to it.

In order to derive this rate, we need a way to argue about distributions across different dimensions and express our generalization bounds only in terms of dimension and number of samples. To this end, we introduce a natural boundedness assumption on the data distribution:

**Assumption 1.** $\|\mathbf{x}\|_2^2 \approx d$ *almost surely.*

We now argue why this is a natural scaling order for this norm: if the entries of the random vector $\mathbf{x}$ are upper bounded as $|\mathbf{x}_i| \leq \alpha_{\max}$ and their average magnitude is lower bounded as $0 < \alpha_{\min} \leq \frac{1}{d} \sum_{i=1}^{d} |\mathbf{x}_i|$. Then

$$\alpha_{\min}\sqrt{d} \leq \frac{1}{\sqrt{d}}\|\mathbf{x}\|_1 \leq \|\mathbf{x}\|_2 \leq \sqrt{d}\|\mathbf{x}\|_\infty \leq \sqrt{d}\alpha_{\max},$$

and hence, Assumption 1 is satisfied. If we think about distributions of pixel (natural image) data of increasing resolution, we indeed have an upper bound on the intensity of each pixel. A lower bounded average pixel intensity only means that images are not arbitrarily dark, which often holds in practice. Any distribution of similar characteristics (e.g., sensor data) will probably satisfy our assumption.

We also introduce a *growth condition on the intrinsic dimension*, which states that $r(\Sigma) \approx d^s$ for some $0 \leq s \leq 1$. Rather than being an assumption, this condition helps to understand how our bounds change as the distribution falls between the two possible extremes given by the bound $1 \leq r(\Sigma) \leq d$. Traditionally, distributions that attain the lower bound ($s = 0$, $r(\Sigma) = 1$) or the upper bound ($s = 1$, $r(\Sigma) = d$) are called *anisotropic* or *isotropic*, respectively.

**Corollary 2.** *Let Assumption 1 hold, and suppose that* $r(\Sigma) \approx d^s$ *for some* $s \in [0,1]$ *and* $n \geq r(\Sigma) \log d$. *Then*

$$\mathcal{R}_{n,\mu}(\mathcal{Q}_{\|\cdot\|_{\mathrm{tr}},\lambda}) \lesssim \lambda \frac{d^{1-s/2}\sqrt{\log d}}{\sqrt{n}} \tag{22}$$

$$\mathcal{R}_{n,\mu}(\mathcal{Q}_{\|\cdot\|_F,\lambda}) \approx \lambda \frac{d}{\sqrt{n}} \tag{23}$$

*Proof.* For the first inequality, note that $\|\mathbf{x}\|_2^2 \approx d$ implies that $\mathrm{tr}\,\Sigma = \mathbb{E}\|\mathbf{x}\|_2^2 \approx d$. Thus, $\frac{\mathrm{tr}\,\Sigma}{\|\Sigma\|_2} = r(\Sigma) \approx d^s \implies \|\Sigma\|_2 \approx d^{1-s}$ Using these two identities in the inequality eq. 3 we obtain the first result.

For the second identity, Lemma 5 implies the class of Frobenius-norm constrained quadratic functions is equal to the ball of radius $\lambda$ in the RKHS corresponding to the quadratic kernel. By Mohri et al. (2018, Theorem 5.5)[1], we have that

$$\hat{\mathcal{R}}(\mathcal{Q}_{\|\cdot\|_F,\lambda}; \mathbf{X}_n) \approx \frac{\lambda}{n}\sqrt{\sum_{i=1}^{n} K(\mathbf{x}_i, \mathbf{x}_i)} = \frac{\lambda}{n}\sqrt{\sum_{i=1}^{n} \|\mathbf{x}_i\|_2^4} \approx \frac{\lambda d}{\sqrt{n}} \tag{24}$$

where the final inequality comes from our assumption that $\|\mathbf{x}\|_2^2 \approx d$. Taking expectation with respect to the sample we obtain that $\mathcal{R}_{n,\mu}(\mathcal{Q}_{\|\cdot\|_F,\lambda}) \approx \lambda\frac{d}{\sqrt{n}}$. $\qquad\square$

Corollary 2 then states that the nuclear norm constraint *adapts* to the intrinsic dimension of the distribution. In the worst case ($s = 0$) it grows linearly[2] with dimension; In the best case ($s = 1$) it is much slower, and grows as the square root of the dimension.

Note that we can bring any distribution to approximate isotropic position, given a good approximation of $\Sigma$. Hence, it makes sense to expect performance gains if we perform such *normalization procedure* before training nuclear-norm constrained quadratic classifiers. In section 5 we will put to test this claim in synthetic datasets, to illustrate the theory.

## 5 Experiments

The results derived in Corollary 2 pertain to the worst case generalization gap. This worst case gap is not guaranteed to be attained by the function found through empirical loss minimization. Therefore, in order to better test our results, we try to find the function that attains this worst case generalization gap.

---

[1]See remarks following the proof
[2]for simplicity we ignore the logarithmic factor

The experiment we propose to illustrate the difference in intrinsic dimension sensitivity consists of computing the quadratic function $f$ such that $\mathbb{E}[\ell(f(\mathbf{x}, y))] - \frac{1}{n} \sum_{i=1}^{n} \ell(f(\mathbf{x}_i, y_i))$ is maximized. The expectation in the term above being generally difficult to compute in closed form, we replace the objective by an empirical estimate

$$\frac{1}{n_{\text{test}}} \sum_{k=1}^{n_{\text{test}}} \ell(f(\mathbf{x}_k, y_k)) - \frac{1}{n_{\text{train}}} \sum_{i=1}^{n_{\text{train}}} \ell(f(\mathbf{x}_i, y_i)).$$

We expect this quantity to more faithfully track the upperbounds derived in Corollary 2. Indeed, what we evaluate here is the capacity of the class constraining $f$ to attain a low loss on $n_{\text{train}}$ samples while attaining a very high one on $n_{\text{test}}$ samples.

To illustrate that Frobenius and nuclear norm constrained classifiers exhibit different behaviors depending on the intrinsic dimension of the data, we compute the result of the maximization procedure on isotropic distributions on one hand and on anisotropic on the other. We set the radius $\lambda = 1$ for both Nuclear and Frobenius norm constrained classifiers. We then observe the evolution of these quantities as the dimension increases.

**Generating Isotropic Data**    To generate isotropic data satisfying our assumptions, we sample a standard Gaussian random vector in $\mathbb{R}^d$ and normalize to obtain i.i.d samples $\mathbf{x}_i$ that are uniformly distributed on the sphere of radius $\sqrt{d}$.

**Generating Anistropic Data**    To generate anisotropic data, we first generate isotropic data features by uniformly sampling points on the sphere of radius $\sqrt{d}$ as described previously. We then transform the samples into more anisotropic ones by multiplying each feature vector coordinate-wise by a vector $\alpha^{(s)}$ defined as $\alpha_i^{(s)} = C_s \frac{1}{i^s}$, $i = 1, \ldots, d$, for $s \in [0, 1]$ where $C_s$ is chosen such that $\|\alpha^{(s)}\|_2 = \sqrt{d}$. That, way, $s = 0$ induces no change on the data, since $\alpha^{(1)} = [1, 1, \ldots, 1]^T$, and larger $s$ implies that the data will be more squashed along the first few coordinates, inducing smaller intrinsic dimension. This transform does not affect the norm of the vectors since $\mathbb{E}[\|\alpha^{(s)} \times \mathbf{x}_i\|^2] = \sum_{k=1}^{d} (\alpha_k^{(s)})^2 \mathbb{E}[[\mathbf{x}_i]_k^2] = \sum_{k=1}^{d} (\alpha_k^{(s)})^2 = \sqrt{d}$

**Generating the labels**    We test two approaches for generating the labels : (1) We generate a random matrix $A$ with i.i.d standard Gaussian coordinates and set $y_i = \text{sgn}(\mathbf{x}_i^\top A \mathbf{x}_i)$, (2) We set the labels randomly with $y_i \sim \text{Bern}(0.5)$.

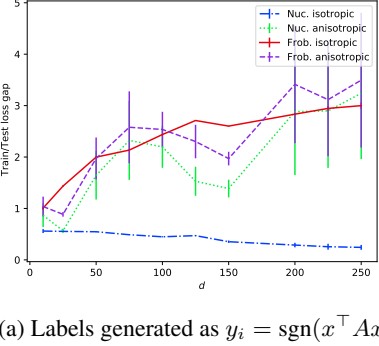
(a) Labels generated as $y_i = \text{sgn}(x^\top A x)$

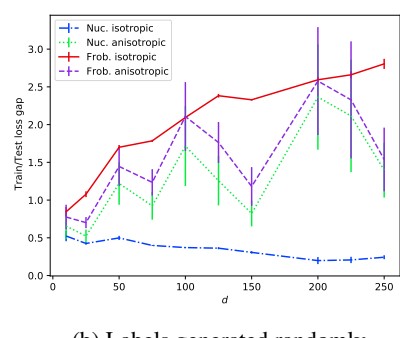
(b) Labels generated randomly

Figure 1: Maximized Train/Test loss gaps for each dimension $d$. The results are averaged over 5 independent runs. The error bars correspond to the standard error.

**The results**    We observe that the maximal gaps do indeed evolve as the theory predicts. We indeed have the nuclear norm constrained classifier on isotropic data not exhibiting the same growth as the others, indicating that the maximal generalization gap for nuclear norm constrained classifiers is indeed sensitive to isotropy. Additional experiments and details are provided in Appendix E.

# 6 Discussion

Our result shows that given a fixed regularization parameter $\lambda$, nuclear-norm constrained classifiers can take advantage of the properties of the data distribution to shave off a $\sqrt{d}$ factor from the Rademacher complexity, whereas Frobenius-norm constrained ones cannot.

We rely on two main elements to show this : Hölder's inequality in equation (5) which introduces the dual norms and a bound on the expected deviations $M_k$, which quantifies how fast the empirical moment estimate converges to the true moment matrix. Both these elements admit extensions to higher order tensors which corresponds to polynomials of higher degree.

The extension of Hölder's inequality is immediate. The main difficulty of extending our result lies in establishing the convergence rate of the deviations $\mathbb{E}[\|\frac{1}{k}\sum_{i=1}^{k} x_i^{\otimes m} - \mathcal{M}\|_{op}]$, where $\mathcal{M} = \mathbb{E}[\mathbf{x}^{\otimes m}]$ is the tensor of $m$-th moments. For matrices, the rate is obtained through the matrix Bernstein inequality. Therefore, an extension can be derived by using the recently proved tensor Bernstein inequality in Luo et al. (2019) obtained by flattening the tensors and recycling the results for matrices.

## Acknowledgements

This work is funded (in part) through a PhD fellowship of the Swiss Data Science Center, a joint venture between EPFL and ETH Zurich. This project has received funding from the European Research Council (ERC) under the European Union's Horizon 2020 research and innovation programme (grant agreement no. 725594 - time-data). This project was sponsored by the Department of the Navy, Office of Naval Research (ONR) under a grant number N62909-17-1-2111. Research was sponsored by the Army Research Office and was accomplished under Grant Number W911NF-19-1-0404. This work was supported by the Swiss National Science Foundation (SNSF) under grant number 200021_178865 / 1.

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
