## A  Appendix

## B  Proof of Lemma 2

For completeness, we include the following result, taken from `https://math.stackexchange.com/questions/3209660/show-that-2m-choose-m-leq-frac22m-sqrt2m`:

**Lemma 6.** *For $m \in \mathbb{N}$ we have*

$$\binom{2m}{m} < \frac{2^{2m}}{\sqrt{2m+1}} < \frac{2^{2m}}{\sqrt{2m}} \tag{25}$$

*Proof.* By induction on $m \in \mathbb{N}$. For $m = 1$ the inequality holds. Suppose it holds for some $m \in \mathbb{N}$

$$\frac{(2m+2)!}{2^{2m+2}((m+1)!)^2} = \frac{(2m)!}{2^{2m}(m!)^2} \cdot \frac{2m+1}{2m+2} < \frac{1}{\sqrt{2m+1}} \cdot \frac{2m+1}{2m+2} \overset{?}{\leq} \frac{1}{\sqrt{2m+3}}. \tag{26}$$

So it remains to prove that

$$4m^2 + 8m + 3 = (\sqrt{2m+1}\sqrt{2m+3})^2 \leq (2m+2)^2 = 4m^2 + 8m + 4 \tag{27}$$

which holds. $\qquad\square$

**Lemma 7.** *For all $n \in \mathbb{N}$ even:*

$$\sum_{k=0}^{n} \binom{n}{k} |2k - n| = n \binom{n}{\frac{n}{2}}$$

*Proof.* Suppose that $n = 2m$ is even.

$$
\begin{aligned}
\sum_{k=0}^{n} \binom{n}{k} |2k - n| &= \sum_{k=0}^{m} \binom{2m}{k} (2m - 2k) + \sum_{k=m+1}^{2m} \binom{2m}{k} (2k - 2m) \\
&= 2\sum_{k=0}^{m} \binom{2m}{k} (m - k) + \sum_{k=0}^{m-1} \binom{2m}{2m-k} (2(2m-k) - 2m) \\
&= 2\sum_{k=0}^{m} \binom{2m}{k} (m - k) + 2\sum_{k=0}^{m-1} \binom{2m}{k} (m - k) \\
&= 4\sum_{k=0}^{m} \binom{2m}{k} (m - k).
\end{aligned}
$$

We now show by induction that $\forall m \in \mathbb{N}$,

$$\sum_{k=0}^{m} \binom{2m}{k} k = m 2^{2m-1}. \tag{28}$$

The result holds trivially true for $m = 0$. Suppose it is true for some $m \geq 0$ and let's prove it remains true for $m + 1$:

$$\sum_{k=0}^{m+1} \binom{2m+2}{k} k = \sum_{k=0}^{m+1} \left( \binom{2m}{k} + 2\binom{2m}{k-1} + \binom{2m}{k-2} \right) k$$

$$= \sum_{k=0}^{m+1} \binom{2m}{k} k + 2 \sum_{k=1}^{m+1} \binom{2m}{k-1} k + \sum_{k=2}^{m+1} \binom{2m}{k-2} k$$

$$= \sum_{k=0}^{m+1} \binom{2m}{k} k + 2 \sum_{k=0}^{m} \binom{2m}{k} (k+1) + \sum_{k=0}^{m-1} \binom{2m}{k} (k+2)$$

$$= 4 \sum_{k=0}^{m} \binom{2m}{k} k + \binom{2m}{m+1} (m+1) + 2 \sum_{k=0}^{m} \binom{2m}{k} - \binom{2m}{m} m + 2 \sum_{k=0}^{m-1} \binom{2m}{k}$$

$$= 4m 2^{2m-1} + 2 \sum_{k=0}^{2m} \binom{2m}{k}$$

$$= m 2^{2(m+1)-1} + 2 \cdot 2^{2m}$$

$$= (m+1) 2^{2(m+1)-1}.$$

We thus proved that eq. 28 is true for all $m \in \mathbb{N}$. Thus, we have:

$$\sum_{k=0}^{m} \binom{2m}{k} (m-k) = m \sum_{k=0}^{m} \binom{2m}{k} - m 2^{2m-1}$$

$$= \frac{m}{2} \left( \sum_{k=0}^{m} \binom{2m}{k} + \sum_{k=m}^{2m} \binom{2m}{k} \right) - m 2^{2m-1}$$

$$= \frac{m}{2} \left( \sum_{k=0}^{2m} \binom{2m}{k} + \binom{2m}{m} \right) - m 2^{2m-1}$$

$$= \frac{m}{2} \left( 2^{2m} + \binom{2m}{m} \right) - m 2^{2m-1}$$

$$= \frac{m}{2} \binom{2m}{m},$$

which concludes the proof. $\qquad\qquad\square$

*Proof of Lemma 2.* Combine Lemma 6 and Lemma 7. $\qquad\qquad\square$

## C  Proof of Theorem 3

**Lemma 8.** *Suppose that $n \gtrsim d(\log(d) + \log(1/\delta))$ and that $\|\mathbf{x}\|_2^2 \lesssim \mathbb{E}\|\mathbf{x}\|_2^2$ almost surely. With probability at least $1 - 2\delta$:*

$$\|\Sigma\|_2 \lesssim \left( 1 - \sqrt{\frac{d(\log d + \log \frac{1}{\delta})}{n}} \right)^{-1} \|\Sigma_n\|_2. \tag{29}$$

*Proof.* The lemma is an application of the matrix Bernstein inequality. Recall that $\Sigma_n = \sum_{i=1}^{n} \mathbf{x}_i \mathbf{x}_i^T$ where $\mathbf{x}_i$ for $i \in \{1, \ldots, n\}$ are independent, identically distributed and, by assumption, satisfy the inequality $\|\mathbf{x}_i\|_2^2 \lesssim \mathbb{E}\|\mathbf{x}_i\|_2^2 = \operatorname{tr}(\Sigma)$ almost surely. Spelling out the definition of $\lesssim$, we have that there exists a $C \geq 1$ such that

$$\forall i \in \{1, \ldots, n\}, \quad \|\mathbf{x}_i \mathbf{x}_i^T\|_2 = \|\mathbf{x}_i\|_2^2 \leq C \operatorname{tr}(\Sigma) \quad \text{almost surely.}$$

Consequently, by matrix Bernstein, for any $t \geq 0$,

$$\mathbb{P}(\|\sum_{i=1}^{n}\left(\mathbf{x}_i\mathbf{x}_i^T - \Sigma\right)\|_2 \geq t) \leq 2d\exp\left(-\frac{\frac{t^2}{2}}{\sigma^2 + K\frac{t}{3}}\right),$$

where $K = C\operatorname{tr}(\Sigma)$ and $\sigma^2 = \|\sum_{i=1}^{n}\mathbb{E}[(\mathbf{x}_i\mathbf{x}_i^T - \Sigma)^2]\|_2$.

Now for any $0 < \delta \leq 1$, we take

$$t = \sqrt{2}\left(\sigma\sqrt{\log d + \log\frac{1}{\delta}} + K(\log d + \log\frac{1}{\delta})\right).$$

The term inside the exponential on the right hand side of the Bernstein bound then becomes

$$-\frac{\frac{t^2}{2}}{\sigma^2 + K\frac{t}{3}} = -\frac{\sigma^2(\log d + \log\frac{1}{\delta}) + 2K\sigma\sqrt{\log d + \log\frac{1}{\delta}}(\log d + \log\frac{1}{\delta}) + K^2(\log d + \log\frac{1}{\delta})^2}{\sigma^2 + \frac{\sqrt{2}K}{3}\sigma\sqrt{\log d + \log\frac{1}{\delta}} + \frac{\sqrt{2}K^2}{3}(\log d + \log\frac{1}{\delta})}$$

$$= -\frac{(\log d + \log\frac{1}{\delta})\left(\sigma^2 + 2K\sigma\sqrt{\log d + \log\frac{1}{\delta}} + K^2(\log d + \log\frac{1}{\delta})\right)}{\sigma^2 + \frac{\sqrt{2}K}{3}\sigma\sqrt{\log d + \log\frac{1}{\delta}} + \frac{\sqrt{2}K^2}{3}(\log d + \log\frac{1}{\delta})}$$

$$\leq -(\log d + \log\frac{1}{\delta}) \qquad (\text{since } \frac{\sqrt{2}}{3} \leq 1 \text{ and } \log d + \log\frac{1}{\delta} \geq 0).$$

Consequently, the event

$$\|\sum_{i=1}^{n}\left(\mathbf{x}_i\mathbf{x}_i^T - \Sigma\right)\|_2 \leq \sqrt{2}\left(\sigma\sqrt{\log d + \log\frac{1}{\delta}} + K(\log d + \log\frac{1}{\delta})\right) \qquad (30)$$

holds with probability at least $1 - 2\delta$. Now, by proceding just like Theorem 5.6.1 of Vershynin (2018), we can bound $\sigma^2$:

$$\sigma^2 = \|\sum_{i=1}^{n}\mathbb{E}[(\mathbf{x}_i\mathbf{x}_i^T - \Sigma)^2]\|_2$$
$$= n\|\mathbb{E}[(\mathbf{x}_1\mathbf{x}_1^T - \Sigma)^2]\|_2$$
$$= n\|\mathbb{E}[(\mathbf{x}_1\mathbf{x}_1^T)^2] - \Sigma^2\|_2 \qquad (\text{by expanding the square})$$
$$\leq n\|\mathbb{E}[(\mathbf{x}_1\mathbf{x}_1^T)^2]\|_2 \qquad (\text{since } \Sigma \succeq 0)$$
$$= n\|\mathbb{E}[\|\mathbf{x}_1\|_2^2(\mathbf{x}_1\mathbf{x}_1^T)]\|_2$$
$$\leq nC\operatorname{tr}(\Sigma)\|\mathbb{E}[\mathbf{x}_1\mathbf{x}_1^T]\|_2 \qquad (\text{by assumption})$$
$$= nC\operatorname{tr}(\Sigma)\|\Sigma\|_2.$$

Plugging this bound into (30) and dividing by $n$, we find that

$$\|\Sigma_n - \Sigma\|_2 \leq \sqrt{2}\left(\sqrt{nC\operatorname{tr}(\Sigma)\|\Sigma\|_2}\frac{\sqrt{\log d + \log\frac{1}{\delta}}}{n} + C\operatorname{tr}(\Sigma)\frac{(\log d + \log\frac{1}{\delta})}{n}\right)$$

Factorizing by $\|\Sigma\|_2$ on the RHS and using the assumption that $Cd(\log d + \log\frac{1}{\delta}) \lesssim n$, we find

$$\|\Sigma_n - \Sigma\|_2 \lesssim \sqrt{\frac{r(\Sigma)(\log d + \log\frac{1}{\delta})}{n}}\|\Sigma\|_2.$$

With this in hand, we simply use the triangle inequality to obtain

$$\|\Sigma\|_2 = \|\Sigma - \Sigma_n + \Sigma_n\|_2$$
$$= \|\Sigma_n - \Sigma\|_2 + \|\Sigma_n\|_2$$
$$\lesssim \sqrt{\frac{r(\Sigma)(\log d + \log\frac{1}{\delta})}{n}}\|\Sigma\|_2 + \|\Sigma_n\|_2$$

Isolating $\|\Sigma\|_2$ on the left hand side, we find that

$$\|\Sigma\|_2 \lesssim \left(1 - \sqrt{\frac{r(\Sigma)(\log d + \log \frac{1}{\delta})}{n}}\right)^{-1} \|\Sigma_n\|_2.$$

Finally we have that $r(\Sigma) \leq d$. □

**Lemma 9.** *Suppose that $\|\mathbf{x}\|_2^2 \leq B$ almost surely. With probability at least $1 - \delta$:*

$$\operatorname{tr} \Sigma \leq \Sigma_n + \sqrt{\frac{B^2 \log(1/\delta)}{2n}} \tag{31}$$

*Proof.* We define

$$\sigma(\mathbf{x}_1, \ldots, \mathbf{x}_n) = \frac{1}{n} \sum_{i=1}^{n} \operatorname{tr}(\mathbf{x}_i \mathbf{x}_i^T) = \operatorname{tr} \Sigma_n \tag{32}$$

The empirical second moment matrix is an unbiased estimator of the second moment matrix $\Sigma = \mathbb{E}\Sigma_n$. Hence

$$\operatorname{tr} \Sigma = \operatorname{tr} \mathbb{E}\Sigma_n = \mathbb{E} \operatorname{tr} \Sigma_n = \mathbb{E}\sigma(\mathbf{x}_1, \ldots, \mathbf{x}_n) \tag{33}$$

We observe that the quantity we want to bound $|\operatorname{tr} \Sigma - \operatorname{tr} \Sigma_n|$ is precisely the deviation of $\sigma$ to its expected value.

In order to obtain a high-probability bound on this deviation, we will make use of McDiarmid's Inequality. We first show that $\sigma$ satisfies the bounded difference inequality

$$|\sigma(\mathbf{x}_1, \ldots, \mathbf{x}_i, \ldots, \mathbf{x}_n) - \sigma(\mathbf{x}_1, \ldots, \hat{\mathbf{x}}_i, \ldots, \mathbf{x}_n)| = \frac{1}{n}|\operatorname{tr}(\mathbf{x}_i \mathbf{x}_i^T - \hat{\mathbf{x}}_i \hat{\mathbf{x}}_i^T)| = \frac{1}{n}|\|\mathbf{x}_i\|^2 - \|\hat{\mathbf{x}}_i\|^2| \leq \frac{B}{n} \tag{34}$$

Hence by McDiarmid's inequality we conclude that

$$P(\operatorname{tr} \Sigma_n - \operatorname{tr} \Sigma < -t) \leq \exp\left(\frac{-2nt^2}{B^2}\right) \tag{35}$$

Letting $\delta = \exp\left(\frac{-2nt^2}{B^2}\right)$ we have $t = \sqrt{\frac{B^2 \log(1/\delta)}{2n}}$ so that with probability at least $1 - \delta$

$$\operatorname{tr} \Sigma \leq \Sigma_n + \sqrt{\frac{B^2 \log(1/\delta)}{2n}} \tag{36}$$

□

*Proof of Theorem 3.* The first result follows after bounding the Rademacher Complexity using Lemma 8 and Lemma 9 in the inequality 18. The probability is obtained with a union bound.

From this result, the final computable uniform generalization bound in eq. 21 follows from eq. 17 and a union bound. □

## D  Proof of Lemma 1 and Lemma 5

*Proof of Lemma 1.* Dividing by $n$ and taking expectation on the following inequality yields the result.

$$\sup_{\|A\| \leq \lambda} \sum_{i=1}^{n} \sigma_i x_i^T A x_i = \sup_{\|A\| \leq \lambda} \|A\| \sum_{i=1}^{n} \sigma_i x_i^T (\|A\|^{-1} A) x_i \leq \lambda \sup_{\|A\| \leq 1} \sum_{i=1}^{n} \sigma_i x_i^T A x_i \tag{37}$$

□

*Proof of Lemma 5.* By the Moore-Aaronszajn Theorem, the RKHS corresponding to the kernel $K(x, y) = \langle x, y \rangle^2$, denoted by $\mathbb{H}$, is built upon the vector space $\mathbb{H}_0$ corresponding to the linear span of functions of the form $K_y = K(\cdot, y) = \langle \cdot, y \rangle^2$ for $y \in \mathbb{R}^d$. We equip $\mathbb{H}_0$ with the inner product:

$$\left\langle \sum_{i=1}^{n} a_i K_{y_i}, \sum_{j=1}^{m} b_j K_{z_j} \right\rangle := \sum_{i=1}^{n} \sum_{i=j}^{m} a_i b_j K(y_i, z_j) \tag{38}$$

$\mathbb{H}$ is precisely the completion of $\mathbb{H}_0$, under the metric induced by this inner product. However, because any $K_y$ is a homogenous polynomial of second degree, $\mathbb{H}_0$ is finite-dimensional, hence closed and $\mathbb{H} = \mathbb{H}_0$. This shows that $\mathbb{H}$ is a subspace of the space of homogenous polynomials of second degree so $f \in \mathbb{H}$ can be represented as $f(x) = x^T A x$ for some symmetric matrix $A$.

On the other hand, for any symmetric matrix we can represent the function $f(x) = x^T A x$ as

$$f(x) = x^T A x = x^T U S U^T x = \sum_{i=1}^d S_{ii} \langle x, u_i \rangle^2 = \sum_{i=1}^d S_{ii} K(x, u_i) \qquad (39)$$

where $A = U S U^T$ is the SVD (orthogonal diagonalization) of the symmetric matrix A, and $u_i$ are the columns of $U$. So any homogeneous polynomial is in the linear span of the functions of the form $K_y = K(\cdot, y)$. We conclude that, as a set, $\mathbb{H}$ is equal to the space of homogeneous polynomials of second degree.

Now we show that its norm is equal to the Frobenius norm of the associated matrix of coefficients. Let $f(x) = x^T A x \in \mathbb{H}$, where $A$ is a symmetric matrix. Let $A = U S U^T$ be the SVD of $A$. We know that $f = \sum_{i=1}^d S_{ii} K_{u_i}$ so that by definition of the RKHS norm

$$
\begin{aligned}
\|f\|_{\mathbb{H}}^2 &= \left\langle \sum_{i=1}^d S_{ii} K_{u_i}, \sum_{i=1}^d S_{ii} K_{u_i} \right\rangle_{\mathbb{H}} = \sum_{i=1}^d \sum_{j=1}^d S_{ii} S_{jj} \langle K_{u_i}, K_{u_j} \rangle \\
&= \sum_{i=1}^d \sum_{j=1}^d S_{ii} S_{jj} K(u_i, u_j) = \sum_{i=1}^d \sum_{j=1}^d S_{ii} S_{jj} \langle u_i, u_j \rangle^2 = \sum_{i=1}^d S_{ii}^2 = \|A\|_F^2
\end{aligned}
\qquad (40)
$$

$\square$

# E Additional experiments

## E.1 A synthetic experiment maximizing the train/test loss gap

The results derived in Corollary 2 pertain to the worst case generalization gap. This worst case gap is not guaranteed to be attained by the function found through empirical loss minimization. Therefore, in order to better test our results, we try to find the function that attains this worst case generalization gap.

The experiment we propose consists of computing the quadratic function $f$ such that

$$\mathbb{E}[\ell(f(\mathbf{x}, y))] - \frac{1}{n} \sum_{i=1}^n \ell(f(\mathbf{x}_i, y_i))$$

is maximized. The expectation in the term above being generally difficult to compute in closed form, we replace the objective by an empirical estimate

$$\frac{1}{n_{\text{test}}} \sum_{k=1}^{n_{\text{test}}} \ell(f(\mathbf{x}_k, y_k)) - \frac{1}{n_{\text{train}}} \sum_{i=1}^{n_{\text{train}}} \ell(f(\mathbf{x}_i, y_i)).$$

We expect this quantity to more faithfully track the upperbounds derived in Corollary 2. Indeed, what we evaluate here is the capacity of the class constraining $f$ to attain a low loss on $n_{\text{train}}$ samples while attaining a very high one on $n_{\text{test}}$ samples.

To illustrate that Frobenius and nuclear norm constrained classifiers exhibit different behaviors depending on the intrinsic dimension of the data, we compute the result of the maximization procedure on isotropic distributions on one hand and on anisotropic on the other. We set the radius $\lambda = 1$ for both Nuclear and Frobenius norm constrained classifiers. We then observe the evolution of these quantities as the dimension increases.

**Generating Isotropic Data**   To generate isotropic data satisfying our assumptions, we sample a standard Gaussian random vector in $\mathbb{R}^d$ and normalize it as follows :

$$\mathbf{x}_i = \sqrt{d} \times \frac{z_i}{\|z_i\|} \qquad \text{where } z_i \sim \mathcal{N}(0, I_d).$$

The $z_i$'s are sampled independently. The procedure generates i.i.d samples $\mathbf{x}_i$ that are uniformly distributed on the sphere of radius $\sqrt{d}$.

**Generating Anistropic Data**  To generate anisotropic data, we first generate isotropic data features by uniformly sampling points on the sphere of radius $\sqrt{d}$ as described previously. We then transform the samples into more anisotropic ones.

The transform consists of multiplying each feature vector coordinate-wise by a vector $\alpha^{(s)}$ defined as $\alpha_i^{(s)} = C_s \frac{1}{i^s}$, $i = 1, \ldots, d$, for $s \in [0, 1]$ where $C_s$ is chosen such that $\|\alpha^{(s)}\|_2 = \sqrt{d}$. That, way, $s = 0$ induces no change on the data, since $\alpha^{(1)} = [1, 1, \ldots, 1]^T$, and larger $s$ implies that the data will be more squashed along the first few coordinates, inducing smaller intrinsic dimension. This transform does not affect the norm of the vectors since

$$\mathbb{E}[\|\alpha^{(s)} \times \mathbf{x}_i\|^2] = \sum_{k=1}^{d} (\alpha_k^{(s)})^2 \mathbb{E}[[\mathbf{x}_i]_k^2] = \sum_{k=1}^{d} (\alpha_k^{(s)})^2 = \sqrt{d}$$

**Generating the labels**  We test two approaches for generating the labels :

1. We generate a random matrix $A$ with i.i.d standard Gaussian coordinates and set $y_i = \text{sgn}(\mathbf{x}_i^\top A \mathbf{x}_i)$.

2. We set the labels randomly with $y_i \sim \text{Bern}(0.5)$.

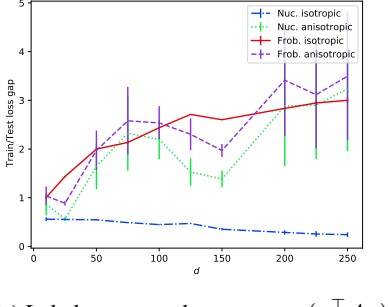
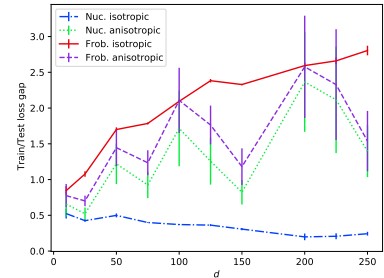

(a) Labels generated as $y_i = \text{sgn}(x^\top A x)$        (b) Labels generated randomly

Figure 2: Maximized Train/Test loss gaps for each dimension $d$. The results are averaged over 5 independent runs. The error bars correspond to the standard error.

**The results**  We observe that the maximal gaps do indeed evolve as the theory predicts. We indeed have the nuclear norm constrained classifier on isotropic data not exhibiting the same growth as the others, indicating that the maximal generalization gap for nuclear norm constrained classifiers is indeed sensitive to isotropy.

## E.2   Additional details on the isotropic normalization effects on real data

Here we give additional details on the experimental setup. As mentioned in the main text, each dataset is randomly split with an 80/20 ratio to define training and testing sets. The training set is used to compute the normalizing factors $\Sigma_n$ and $\nu_n$. These factors are then used to normalize the entire dataset.

The nuclear- and Frobenius norm (or SVM) classifiers each have a hyperparameter $C$ and $\lambda$ that needs to be set before training. We search for the optimal hyperparameter among the grids [0.0000001, 0.000001, 0.00001, 0.0001, 0.001, 0.01, 0.1, 1, 10] for $C$, and [0.0001, 0.001, 0.01, 0.1, 1, 10, 50, 100, 500, 1000, 10000] for $\lambda$. The best hyperparameter is determined through 4-fold cross-validation on the training set.

The SVM classifier results are computed using scikit learn's Buitinck et al. (2013)[3] liblinearFan et al. (2008)[4] wrapper. Details on the nuclear norm classifier are given below.

### E.3 Computational details on the Nuclear Norm Constrained Classifier

The classifier solves the following optimization problem

$$\min_{A \in \mathbb{R}^{d \times d}} \quad \frac{1}{n} \sum_{i=1}^{n} \ell(\mathbf{x}_i^\top A \mathbf{x}_i, y_i)$$

$$\text{subject to} \quad \|A\|_{nuc} \leq \lambda$$

using accelerated projected gradient descent. The projection step makes use of the simplex projection algorithm proposed in Duchi et al. (2008)[5]. The loss function used is the smoothed hinge loss, and its smoothness constant is upper bounded by $L = \frac{1}{n} \sum_{i=1}^{n} \|\mathbf{x}_i\|_2^4$. We use $1/L$ as the stepsize for the accelerated projected gradient descent. The gradients are computed with JAX Bradbury et al. (2018)[6]. For the real dataset experiments, additional variables $b \in \mathbb{R}^n$ and $c \in \mathbb{R}$ are added in order to optimize over non-homogeneous quadratic polynomials like the SVM classifier.

---

[3]Buitinck, L., Louppe, G., Blondel, M., Pedregosa, F.,Mueller, A., Grisel, O., Niculae, V., Prettenhofer, P.,Gramfort, A., Grobler, J., Layton, R., VanderPlas, J.,Joly, A., Holt, B., and Varoquaux, G. API design for ma-chine learning software: experiences from the scikit-learnproject. InECML PKDD Workshop: Languages for DataMining and Machine Learning, pp. 108–122, 2013.

[4]Fan, R.-E., Chang, K.-W., Hsieh, C.-J., Wang, X.-R., andLin, C.-J. Liblinear: A library for large linear classifi-cation.J. Mach. Learn. Res., 9:1871–1874, June 2008.ISSN 1532-4435.

[5]Duchi, J., Shalev-Shwartz, S., Singer, Y., and Chandra, T.Efficient projections onto the l 1-ball for learning in highdimensions. InProceedings of the 25th internationalconference on Machine learning, pp. 272–279, 2008

[6]Bradbury, J., Frostig, R., Hawkins, P., Johnson, M. J., Leary,C., Maclaurin, D., Necula, G., Paszke, A., VanderPlas, J.,Wanderman-Milne, S., and Zhang, Q. JAX: composabletransformations of Python+NumPy programs, 2018. URLhttp://github.com/google/jax.