# OpenReview forum: "The Effect of the Intrinsic Dimension on the Generalization of Quadratic Classifiers"
_NeurIPS.cc/2021/Conference — NeurIPS 2021 Poster_

### Official Review · Reviewer_fnjC · 2021-07-13

**Rating:** 7
**Confidence:** 4

**Summary:**

The paper studies the generalization ability of quadratic forms under trace norm regularization.
The Rademacher complexity of the trace-norm constrained ball is bounded in dependence on the intrinsic dimension (ID) of the data, it is argued that the bound so obtained improves over existing bounds by a factor of sqrt(ID), and it is concuded, that the bound so obtained shows an advantage for isotropic distributions, thus justifying whitening of the data. Experimental evidence is provided. There is also a section devoted to data-dependent bounds.


**Limitations And Societal Impact:**

adequately adressed

**Main Review:**

Overall I liked the paper.
The proof of Theorem 1 (the bound on the Rademacher complexity) is elegant and clear and has some interesting ideas, and the conclusions drawn from it are convincing. The comparison to other bounds suffers a bit from the fact that trace-norm-regularization has not been much studied for the classes defined by matrices as quadratic forms, but rather for matrices operating on matrices, so the motivation is different.
The work seems still very relevant, one thinks of the quadratic kernels from the earlier times of kernels in machine learning. The comparison of trace-norm and Frobenius-norm constraints in section 4 is illuminating.

I didn't like the section on computable generalization bounds. For one reason "computable" seems an exaggeration in a paper which consistently avoids explicit constants. Also the proofs use by-now-standard techniques to estimate trace and spectral norm of the second moment matrix from the data. This section does not contribute very much.
Instead the authors could have clarified the connection to the role of intrinsic dimension in neural network learning, alluded to in the introduction. Statements like "The study of quadratic classifiers can serve as an important first step in understanding how neural networks take advantage of the intrinsic dimension to learn with fewer samples" generate a curiosity, which remains frustrated.



**Time Spent Reviewing:**

6

---

> ### Author Response · Authors · 2021-08-09
> **Authors' Response**
>
> Thank you for taking the time to review our submission. We appreciate your comments on the comparison of the trace-norm and the Frobenius-norm.
>
> We included the section on computability for completeness. It can be moved to the appendix.
>
> The links to neural network learning are indeed interesting. As noted in our response to reviewer WNca, our work points towards the following avenues of research. The most direct application of our results is on networks with quadratic activations or multiplicative interactions [A]. The second more interesting avenue is to show that adaptivity of neural networks to the data might be due to small nuclear norm of the weight matrices. The implicit regularization effects of SGD, which, at least for linear neural networks, is argued to be a form of rank minimization, is closer to a trace-norm regularization than a Frobenius-norm one because trace-norm minimization is a better proxy of rank minimization. Consequently, it may be this implicit regularization that explains the data adaptivity. These speculations however require careful analysis and we judged it best suited for follow-up research.
>
> We will nonetheless add a few lines detailing the following points. First, we can describe in more detail the empirical observations that studied intrinsic dimension and its influence on generalization. Second, we can write out the derivation showing that a single hidden layer neural network with quadratic activations \\(x \mapsto x^2\\) is exactly a non-convex parametrization of a quadratic classifier. We believe this will clarify our position that our work serves as an important first step in theoretically characterizing why neural networks generalize well.
>
> Our main message, however, remains that a simple model can exhibit some of the observed phenomena in complicated, difficult-to-analyze settings. Regularization can explain improved sample complexity over kernels.
>
>
> # **References**
> [A] MULTIPLICATIVE INTERACTIONS AND WHERE TO FIND THEM
> Siddhant M. Jayakumar, Wojciech M. Czarnecki, Jacob Menick, Jonathan Schwarz, Jack Rae, Simon Osidnero, Yee Whye Teh, Tim Harley, Razvan Pascanu (DeepMind). ICLR 2020.

---

### Official Review · Reviewer_DAu4 · 2021-07-14

**Rating:** 7
**Confidence:** 3

**Summary:**

This paper derives generalization bounds using Rademacher complexity for quadratic classifiers, with a nuclear norm constraint on the coefficient matrix. The generalization bound depends on the “intrinsic dimension,” which is a measure of the isotropy of the input distribution, and the largest eigenvalue of the second moment matrix of the distribution. A high-probability computable generalization bound is also given. The authors note that the nuclear norm setting is sensitive to the intrinsic dimension, while the Frobenius norm setting is not. This observation is corroborated by experiments.

**Main Review:**

The results are interesting and relevant, with insightful comparison between nuclear norm and Frobenius norm constraints. The results also seem relevant to understanding how neural networks learn from few samples, as explained by the authors.

My main concern is with the style of presentation. Key terms (e.g. isotropic) are defined rather late. Similarly, models are not presented in detail (what exactly is the quadratic classifier?) until quite late.  Concepts are sometimes used without introduction (e.g. the “pixel space” on pg. 1). Please carefully check the order in which you present the content.

Typos: “triangle inequality” missing the (pg. 4), of vs off (pg. 10), Holder missing accent (pg. 10)


**Time Spent Reviewing:**

1.5

---

> ### Author Response · Authors · 2021-08-09
> **Authors' Response**
>
> Thank you for taking the time to review our paper. We will make the following changes to improve the presentation.
>
> We will precisely define quadratic classifiers as classifiers based on the sign of a quadratic polynomial. We will add that by pixel space, we mean the space \\([0, 1]^d\\) where images belong, as they are arrays of bounded values.
>
> We will also make sure that the definition of _isotropicity_ appears before the term is mentioned in the abstract, precisely, we will rewrite the first sentence as  “It has been recently observed that neural networks, unlike kernel methods, enjoy a reduced sample complexity when the distribution is isotropic (i.e., when the covariance matrix is the identity).”
>
> Finally, we will correct the typos that you mentioned.

---

### Official Review · Reviewer_WNca · 2021-08-03

**Rating:** 6
**Confidence:** 3

**Summary:**

This paper computes Rademacher complexity estimates for regularised quadratic function classes, and shows that the choice of nuclear norm regularisation makes the complexity to scale with the effective dimension of the covariance (trace / largest eigenvalue) of the underlying data distribution. Numerical experiments are also given to demonstrate this effect and show that Frobenius norm regularisation behaves differently. The results are used to bound the generalisation error of quadratic classifiers.

**Limitations And Societal Impact:**

The authors did not explicitly mention limitations.

**Main Review:**

The findings are interesting, albeit the work doesn't seem particularly challenging. The authors plan to extend the analysis to higher order polynomials, which will be an interesting addition.

It is not entirely clear what is the practical implication of the findings so far. Should it be a good idea to use nuclear norm regularisation rather than Frobenius norm in conjunction with quadratic classifiers? (or perhaps more generally?) How was \lambda set in the experiments?

I feel there is a seed of a good idea here albeit the work seems to be in early stages.

The assumption \|x\|_2^2 \ge const E\|x\|_2^2 almost surely seems strange to me. Can the authors justify it?

I'm not sure how to interpret the experiments. Are we looking for the worst function in the class? Also, in Fig 1 (b) the labels are randomly assigned, but the generalisation gap seems to shrink as d increases?

The writing quality can be improved in many places.
- In Proof of Cor 2 it says "both identities in eq. 3" -- but eq(3) is one inequality!
- bottom of p 8: "try to find" sounds vague - indeed, what if the optimiser doesn't find the function we look for?
- top of p 10: what is mean by "an equal regularisation parameter"?

Post rebuttal:
The authors addressed all clarity issues.
I don't think the setting of $\lambda=1$ and the setting of this same value for all regularizers is justified - why would this be a good value? I can see that it doesn't change the conclusion of the paper, but still a good justification would help, otherwise this might send the wrong message to practitioners or to beginners in the field.

On another note, the authors might also be interested in the rank minimising behaviour of the nuclear norm beyond the quadratic classifier, as it was found previously e.g. in matrix completion problems:
Benjamin Recht. A Simpler Approach to Matrix Completion, Journal of Machine Learning Research 12 (2011) 3413-3430.



**Time Spent Reviewing:**

20

---

> ### Author Response · Authors · 2021-08-09
> **Authors' Response**
>
> Thank you for taking the time to review our submission. Here are some clarifications regarding your observations. We hope that they will improve your opinion of our work and we kindly ask you to consider the possibility of raising your score.
>
> * **Regarding the practical implications of our work:** Although the main purpose and goal of our analysis is to highlight the different generalization behaviors of different regularizers, it is possible to draw some practical implications from our work: it might be a good idea to use nuclear norm rather than Frobenius norm regularisation in conjunction with quadratic classifiers, especially in high dimensions when the data is nearly isotropic. However, we stress that our paper is not arguing for the use of a certain regularizer over the other, we are simply showing that an observed phenomenon (improved sample complexity) can be explained by regularization.
>
>     More generally, our results also *suggest* (but not imply) two more promising avenues of research: (1) it might be a good idea to try nuclear norm regularization in multiplicative networks [A] which are composed of quadratic layers and (2) adaptivity of neural networks to the data might be due to small nuclear norm of the weight matrices (this aligns nicely with the line of work arguing that SGD is implicitely minimizing rank, for which the nuclear norm is a proxy). Of course, **these suggestions are not part of the contributions of our paper**. We will clarify this in the conclusion section in the final version.
>
> * **Regarding the choice of \\(\lambda\\) in the experiments:** We set lambda to 1 in the experiments. (We have indicated it on line 307, but we can make it figure more prominently)
> * **Regarding the assumption \\(\|x\|_2^2 \geq c \mathbb{E} \|x\|_2^2 \\) almost surely:** We think you mean the inequality \\(\|x\|_2^2 \leq c \mathbb{E} \|x\|_2^2\\) in Lemma 4 (line 207)? This is only a technical assumption which means that the distribution is bounded, and that its magnitude concentrates near the mean. This is in particular satisfied for natural images with fixed average pixel magnitude. By writing it this way, we are considering the entire class of bounded distributions whose magnitudes are allowed to scale with dimension, but once normalized by the expected norm, the dimension dependence dissappears. Indeed without such assumption, the empirical covariance is not a good estimator of the true covariance (see thm 5.6.1 and ex 5.6.5 in [B]).
> * **Regarding the interpretation of the experiments, and Figure 1.b**: Our theoretical bounds, as all uniform generalisation bounds, are on the worst function in the class so we designed the experiment to find this worst performing function. We do so to exhibit the intrinsic dimension dependence of the worst train-test gap and corroborate our theoretical findings.
>
>     We think that the observed decrease in Figure 1.b is not substantial enough to conclude that there is a downward trend, for instance, in the rightmost part of the plot, we see that the generalization gap starts to increase slightly, breaking the downwards trajectory. Overall, we believe the trend is *flat*. The main message is that there is clearly a _different sensitivity_ to an increase in dimension between the different regularizations. The plot is a result of a non-convex optimization  in increasing dimensions, so some variability is to be expected.
> * **Regarding the writing syle**:
>   * *<u>In Proof of Cor 2 it says "both identities in eq. 3" -- but eq(3) is one inequality!</u>*: We will rephrase this in a clearer way. What we mean is that one should use the two identities \\(r(\Sigma) \approx d^s \\) and \\( \|\Sigma\|_2 \approx d^{1-s}\\) together with the inequality given in (3), to arrive at the final result. This line should be read as “Plugging the two previously mentioned identities in the right-hand side of inequality (3)...”
>   * *<u>bottom of p 8: "try to find" sounds vague - indeed, what if the optimiser doesn't find the function we look for?</u>*: The problem is a non-convex maximization problem, therefore, if the optimiser does not find the maximum, it nevertheless provides a lower bound.
>   * *<u>top of p 10: what is mean by "an equal regularisation parameter"?</u>*: This line means that the parameter \\(\lambda\\) has the same numerical value for both regularizations.
>
>
> # **References**
> [A] MULTIPLICATIVE INTERACTIONS AND WHERE TO FIND THEM
> Siddhant M. Jayakumar, Wojciech M. Czarnecki, Jacob Menick, Jonathan Schwarz, Jack Rae, Simon Osidnero, Yee Whye Teh, Tim Harley, Razvan Pascanu (DeepMind). ICLR 2020.
> [B] High-Dimensional Probability: An Introduction with Applications in Data Science. Roman Vershynin. 2020. https://www.math.uci.edu/~rvershyn/papers/HDP-book/HDP-book.html

---

### Decision · Program_Chairs · 2021-09-27

**Decision:**

Accept (Poster)

**Comment:**

The paper presents an interesting observation on the effect of different norms used
for regularizing quadratic classifiers. In short, it presents an upper bound on the Rademacher
averages of quadratic classifiers with bounded nuclear norm that nicely includes effects
such as the intrinsic dimension of the data measured in terms of norms of the covariance matrix.
While it is not entirely surprising that different regularizing norms may have a different
dependence on this intrinsic dimensionality, the paper provides solid and interesting results.
On the downside, the wider significance of the considered class of classifiers is at least
unclear, and the rebuttal phase has not sufficiently resolved this problem. Compared to
other papers in my badge, I am thus somewhat skeptical in view of impact.